# Genotyping Analysis of *Cryptococcus deuterogattii* and Correlation with Virulence Factors and Antifungal Susceptibility by the Clinical and Laboratory Standards Institute and the European Committee on Antifungal Susceptibility Testing Methods

**DOI:** 10.3390/jof9090889

**Published:** 2023-08-31

**Authors:** Leonardo Euripedes Andrade-Silva, Anderson Vilas-Boas, Kennio Ferreira-Paim, Juliana Andrade-Silva, Daniel de Assis Santos, Thatiana Bragine Ferreira, Aercio Sebastião Borges, Delio Jose Mora, Marcia de Souza Carvalho Melhem, Mario Léon Silva-Vergara

**Affiliations:** 1Infectious Diseases Unit, Internal Medicine Department, Federal University of Triangulo Mineiro, Uberaba 38001-170, MG, Brazil; anderson.vilasboas@gmail.com (A.V.-B.); kenniopaim@gmail.com (K.F.-P.); julianaeandrade02@gmail.com (J.A.-S.); tatibragin@yahoo.com.br (T.B.F.); marioleon1956@gmail.com (M.L.S.-V.); 2Microbiology Department, Biological Sciences Institute, Federal University of Minas Gerais, Belo Horizonte 31270-901, MG, Brazil; dasufmg@gmail.com; 3Infectious Diseases Unit, Internal Medicine Department, Federal University of Uberlândia, Uberlândia 38496-017, MG, Brazil; 4Center of Health Sciences, Federal University of Sul da Bahia, Teixeira de Freitas 85866-000, BA, Brazil; delioj@gmail.com; 5Mycology Department, Instituto Adolfo Lutz, São Paulo 01246-902, SP, Brazil; melhemmr@uol.com.br

**Keywords:** *Cryptococcus gattii* species complex (CGSC), *Cryptococcus deuterogattii*, CLSI, EUCAST, virulence factors

## Abstract

Data about the relationship between their molecular types, virulence factors, clinical presentation, antifungal susceptibility profile, and outcome are still limited for *Cryptococcus deuterogattii*. This study aimed to evaluate the molecular and phenotypic characteristics of 24 *C*. *deuterogattii* isolates from the southeast region of Brazil. The molecular characterization was performed by multilocus sequence typing (MLST). The antifungal susceptibility profile was obtained according to CLSI-M27-A3 and EUCAST-EDef 7.1 methods. The virulence factors were evaluated using classic techniques. The isolates were divided into four populations. The molecular analysis suggests recombinant events in most of the groups evaluated. Resistance and susceptibility dose-dependent to fluconazole were evidenced in four isolates (16%) by EUCAST and in four isolates (16%) by CLSI methods. The agreement at ±two dilutions for both methods was 100% for itraconazole, ketoconazole, and voriconazole, 96% for amphotericin B, and 92% for fluconazole. Significant differences in virulence factor expression and antifungal susceptibility to itraconazole and amphotericin B were found. The mixed infection could be suggested by the presence of variable sequence types, differences in virulence factor production, and decreased antifungal susceptibility in two isolates from the same patient. The data presented herein corroborate previous reports about the molecular diversity of *C*. *deuterogattii* around the world.

## 1. Introduction

Cryptococcosis is a global fungal infection that occurs predominantly in immunocompromised hosts or apparently immune-competent individuals. It is caused by the *Cryptococcus neoformans* species complex (CNSC) and the *Cryptococcus gattii* species complex (CGSC), respectively [1]. The clinical presentation, outcome, geographical distribution, host preference, ecological niche, and molecular profile are different in both species.

Until recently, four major molecular types were recognized in *C. gattii* isolates recovered from different places around the world [2,3,4]. Recently, a new taxonomical view proposed that each major molecular type of *C. gattii* would receive a species status, e.g., VGI—*Cryptococcus gattii*, VGII—*Cryptococcus deuterogattii*, VGIII—*Cryptococcus bacillisporus*, and VGIV—*Cryptococcus tetragattii* [5]. Nevertheless, this proposal remains under discussion [5,6,7]. Furthermore, the *C. gattii* species complex (CGSC) diversity spectrum is ongoing, as indicated by the recent discovery of VGV and VGVI genotypes in environmental isolates from Africa [8] and by the increase in the number of data confirming *C. decagatti* as VGVI [8,9,10].

*Cryptococcus gattii* and *C*. *deuterogattii* (serotypes B and C) usually affect apparently immunocompetent hosts, whereas *C. bacillisporus* and *C. tetragattii* have been commonly isolated from immunocompromised patients [11,12,13]. The isolates of the VGV lineage were recovered from Zambian woodland environments in association with the nitrogen-rich middens of a small mammal known as Hyrax [8]. Otherwise, *C. decagatti* has already been obtained from clinical cases and a veterinary isolate [8,10].

The species *C. gattii*, *C. deuterogattii*, *C. bacillisporus,* and *C. tetragattii* can be found globally, but the *C. gattii* and *C. deuterogattii* genotypes are the most predominant [2].

The relevance of *C. deuterogattii* was highlighted in 1999 when it was identified as the agent related to the Vancouver Island outbreak of cryptococcosis. Later, it expanded to other geographical regions of North America, which pointed out the ongoing geographical expansion of this species, formerly restricted to tropical and subtropical areas [2,14,15,16,17].

Several virulence factors of the CNSC and CGSC complexes have been previously documented. These include the polysaccharide capsule, melanin production, urease, extracellular enzymatic activities, and the ability to grow at a temperature of 37 °C, among others [1,18,19,20]. Although the pathogenic *Cryptococcus* species express all or most of these factors, the variability in their expression would explain in part the greater virulence of *C. deuterogattii* isolates. Despite, the relationship between the expression of the virulence factors and the genetic profile is still unclear [21,22].

Differences in antifungal susceptibility profiles and virulence factor production among the main molecular types of CGSC have already been reported [23,24,25,26]. Recent studies pointed out that these differences are related to the group I introns in the mitochondrial large subunit rRNA gene (LSU) [27] and the significantly higher expression of ABC transporters [28]. Nevertheless, it is unclear whether isolates of the same genotype can also present differences in their virulence factors’ expression [21].

Currently, in vitro antifungal susceptibility testing in *Cryptococcus* spp. can be carried out via standardized broth microdilution methods promulgated by both the Clinical and Laboratory Standards Institute [CLSI] [29,30] and The European Committee on Antifungal Susceptibility Testing [EUCAST] [31], among others.

Recently, the frequency of *C. deuterogattii* isolates presenting resistance or a non-wild-type profile to azoles and amphotericin B exhibited variability from less than 1% in Taiwan [32] and in a study that included different species of CGSC from different continents [33] to 80% in Brazil [34]. In Brazil, resistance and/or non-wild-type *C. deuterogattii* isolates have already been described for different antifungals. Such a phenotype was described as fluconazole in clinical isolates, with different frequencies ranging from the absence of this phenotype [35], 25.5% [36], to 80% [34]. Such a phenotype has also been described for other azoles, such as itraconazole [37] and voriconazole [37,38].

The aim of this study was to compare the genetic profile of *C. deuterogattii* isolates from a non-endemic area of Brazil with their phenotypic characteristics.

## 2. Materials and Methods

### 2.1. Isolates and Main Clinical Information

A total of 24 *C. deuterogattii* isolates from the southeast region of Brazil were included (23 clinical and 1 environmental). These isolates had been previously genotyped by Multilocus sequencing typing (MLST), and the main data about clinical information, genotypic results, and biological results are described in Appendix A and have already been published elsewhere [39].

### 2.2. Determination of Mating Types

Genomic DNA extraction and mating type PCR of these isolates were performed in accordance with previous report [40].

### 2.3. Phylogenetic Analysis

The phylogenetic analysis was performed in MEGA 6.0 [41]. The consensus sequences of the isolates were aligned with the Clustal W2 algorithm (https://www.ebi.ac.uk/Tools/msa/clustalw2/, accessed on 15 April 2022) [42]. The allelic sequences for each isolate were concatenated, and the evolutionary relationships, with 1000 bootstrap replicates, were inferred using maximum likelihood (ML), neighbor-joining (NJ), and unweighted pair group method with arithmetic mean (UPGMA) methods [41].

### 2.4. Minimum Spanning Tree, Multidimensional Scaling Plots, and Split Decomposition Analysis

The minimum spanning tree was generated using the goeBURST algorithm in the PHILOVIZ II software (http://www.phyloviz.net/wiki/, accessed on 15 April 2022) [43,44]. This analysis was generated from the concatenated sequence regions to visualize the relatedness of geographic origin, production of virulence factors, and antifungal susceptibility profiles with their allelic profiles.

Classical Multidimensional Scaling plots were constructed using a similarity matrix by arranging samples in two-dimensional space according to their relative similarities in the software MLSTest v.1.0.1.23 [45]. In addition, the split decomposition analysis was performed in the software SplitsTree v. 4.13.1 [46] to evaluate the distribution of the main groups found by the other molecular techniques.

### 2.5. Nucleotide Diversity

The software DNAsp 5.10 [47] was used to calculate the extent of the main markers of DNA polymorphisms. In addition, the presence of recombination was also checked in the software SplitsTree v. 4.13.1 [46].

### 2.6. Genetic Differentiation Based on Sequences of Populations

A hierarchical analysis of molecular variance (AMOVA) was performed in Arlequin 3.1 in order to examine the distribution of genetic variation and determine the extent of connectivity among populations based on allelic profiles. Statistical significance was determined by comparing the observed results with 10,000 permutated datasets established with a null hypothesis of no genetic differentiations among geographic populations within each analyzed dataset. The population differentiation test (Fst) was calculated from concatenated sequences of the seven MLST housekeeping genes and used to visualize the genetic distance among populations evaluated in the present study.

### 2.7. Virulence Factors and Antifungal Susceptibility

The ability of capsular synthesis was measured as previously described [48]. The main virulence factors—phospholipases, proteases, gelatine hydrolysis, and hemolytic activity—were evaluated by the Pz in accordance with previous reports [49,50,51,52,53]. The Pz value was interpreted as follows: Pz = 1.0, negative activity; Pz = 0.7 to 0.99, low activity; Pz = 0.5 to 0.69, moderate activity; and Pz < 0.5, high activity [54].

The melanin production was evaluated by direct visualization and spectrophotometry [55]. The results were reported as optical densities (OD) at 480 nm and represented by the arithmetic mean absorbance values [56,57]. The result was interpreted as follows: OD < 0.45, low activity; OD = 0.45 to 0.69, moderate activity; and OD > 0.69, high activity. The urease activity was evaluated by spectrophotometry in urea Christensen broth (Difco, Sparks, MD, USA) [58]. The result was interpreted as follows: OD < 1.1, low activity; OD = 1.1 to 1.3, moderate activity; and OD > 1.3, high activity.

All experiments were performed in quadruplicate with at least two independent replications, and the results were described by the generated average. In all experiments, the reference strains *C. gattii* ATCC 24065 (serotype B) and *Candida krusei* ATCC 6258 were used as positive and negative controls, respectively.

The antifungal susceptibility tests were performed using the broth microdilution technique following the Clinical and Laboratory Standards Institute (CLSI) recommendations available in the documents M27-A3 [29], Supplement 4 of CLSI [30], and European Committee for Antimicrobial Susceptibility Testing—EUCAST-EDef 7.1 [31]. The optical density (DO) values were recorded on the microplate reader (Benchmark Plus, Bio-Rad^®,^, Alfred Nobel Drive, Hercules, CA, USA).

The drugs tested by the two methodologies were amphotericin B (AMB) (Bristol-Myers Squibb, Princeton, NJ, USA), itraconazole (ITZ) (Janssen Pharmaceuticals, Beerse, Belgium), voriconazole (VRZ) (Pfizer, New York, NY, USA), fluconazole (FLZ) (Pfizer), and ketoconazole (KTZ) (Janssen Pharmaceuticals). Due to the current absence of standard clinical breakpoints for *Cryptococcus* spp., the epidemiological cut-off values (ECVs) herein used were those described on the guidelines of the CLSI for *C. gattii* VGII as follows: AMB 1.0 μg/mL; ITZ 1.0 μg/mL; VRZ 0.5 μg/mL; and FLZ 32.0 μg/mL [30]. The breakpoints for KTZ were defined as ≥2.0 μg/mL [59]. Also, as there are currently no standard clinical breakpoints defined for *Cryptococcus* under the EUCAST method, the breakpoints used were those described in the document breakpoint table for interpretation of MICs Version 10.0 for *Candida* species with normal susceptibility to fluconazole. These breakpoints are as follows: susceptible ≤ 2.0 μg/mL, intermediate 2.0–4.0 μg/mL, and resistance > 4.0 μg/mL (EUCAST Antifungal Clinical Breakpoint Table v. 10.0).

### 2.8. Statistical Analyses

Statistical analyses of phenotypic and molecular data were performed using Bioestate v. 5.0 (https://www.mamiraua.org.br/pt-br, accessed on 10 April 2021), MS Excel (Microsoft Corporation, Redmond, WA, USA), and GraphPad PRISM v. 6.0 (https://www.graphpad.com, accessed on 10 April 2021). The normality of the data was evaluated using the D’Agostino Pearson test. The homogeneity of variances among groups was tested by Bartlett’s test when the data presented a normal distribution. After these tests, all phenotypic and clinical data analyses were performed by non-parametric tests. The variables were evaluated by the Mann–Whitney test to compare two groups and the Kruskal–Wallis test to compare three or more groups, applying Dunn’s post-test if necessary. The correlation between the two variables was evaluated through the Spearman test. To compare the genetic data with categorical variables, the Fisher’s exact test or chi-square test was used. *p*-values less than 5% (*p* < 0.05) were considered statistically significant.

## 3. Results

### 3.1. MLST, Population Structure, and Genetic Variability

The 24 *C. deuterogattii* isolates herein evaluated presented matting type α, 24 different STs, and were grouped in four different populations (e.g., Pop1, Pop2, Pop3, and Pop4). Pop1 to Pop3 was demonstrated by different molecular analyses such as phylogenetic analysis using ML, NL, and UPGMA methods (Figure 1A–C), classical multiscaling analysis (Figure 1D), and split decomposition analysis (Figure 1E). The pop4 was genetically more diverse and did show a heterogeneous molecular group. The Pop1 was smaller and composed of three clinical isolates (ST346-G10, ST340-G04, and ST345-G09) from the Triângulo Mineiro region (TM). The Pop2 was composed of 12 isolates from TM and São Paulo (SP). The Pop3 was composed of five isolates from the TM region (ST339-G3, ST343-G7, and ST347-G11), Belo Horizonte (BH) (ST354-G18), and SP (ST361-G25). The Pop4 included three isolates from BH (ST350-G14, ST351-G15, and ST355-G19) and one (ST357-G21) from SP.

The sequence analysis of these isolates exhibited high variability (Hd = 1.00, π = 0.00144–0.03192, and k = 6.000–131.33) (Table 1). The most polymorphic was the Pop4 (Hd = 1.00, π = 0.03192, and k = 131.33), and the least polymorphic was the Pop1 (Hd = 1.00, π = 0.00144, and k = 6.000). In accordance with their origin, the most polymorphic group was composed of isolates from BH (Hd = 1.00, π = 0.02070, and k = 85.400), and the least polymorphic group was composed of isolates from the TM region (Hd = 1.00, π = 0.00317, and k = 13.179).

The values of Fst revealed significant molecular differences among populations evaluated in the present study (Fst ranging from 0.023 to 0.599) (Table 2). The populations most closely related were Pop3 × Pop4 (Fst = 0.126), followed by Pop3 × Pop2 (Fst = 0.249), and the more distant were Pop1 × Pop3 (Fst = 0.599), followed by Pop1 × Pop2 (Fst = 0.493). In accordance with their clinical site, the most polymorphic group included those isolates recovered from cerebrospinal fluid (Hd = 1.00, π = 0.03192, and k = 14.267), and the least polymorphic group included isolates from the skin (Hd = 1.00, π = 0.00193, and k = 8.000). The neutrality tests (Tajima’s D, Fu and Li’s D, Fu and Li’s F, and Fu’s Fs) evidenced purifying selection or population expansion for most evaluated groups. The results obtained by the Watterson estimator (theta) method and by the PHI test suggest recombinant events in most of them (Table 1).

The isolates G8 (ST344) and G9 (ST345) were recovered from skin fragments of an apparently immunocompetent patient in two different cultures with a 22-day interval between them. These isolates showed differences in the *GPD1* and *SOD1* loci. In the *GPD1* locus, the G8 isolate showed AT21, while the G9 isolate presented AT6. The nucleotide difference between these ATs was based on the insertion of guanine in AT21 at position 157 in relation to AT6. In the *SOD1* locus, the G8 isolate showed AT14, while the G9 isolate presented AT58. These ATs exhibited several nucleotide differences, as described in Table 3. Moreover, the isolates also showed phenotypic differences in virulence factor production, and G9 presented a mildly decreasing antifungal susceptibility to fluconazole by CCLI and EUCAST methods (Appendix A and Figure 2).

### 3.2. Virulence Factors

The capacity for capsular synthesis was verified in all *C. deuterogattii* isolates. No differences in the production of melanin, proteases, yeast size, hemolytic activity, proteases, or phospholipases were found. A positive correlation between urease activity with hemolytic activity and gelatin hydrolysis with phospholipases was observed (Appendix A). A negative correlation between urease activity with phospholipase and urease activity with gelatin hydrolysis was identified (Appendix A).

Differences in urease activity among Pop1 (1.513 ± 0.15) × Pop2 (1.135 ± 0.11) and in Pop4 (1.47 ± 0.14) × Pop2 (1.135 ± 0.11) (*p* = 0.0042) were found (Figure 3B). Hemolytic activity was evident in most isolates. The Pop4 isolates exhibited high or medium activity, and there was a difference between Pop1 (0.93 ± 0.11) and Pop4 (0.39 ± 0.08) (*p* = 0.015) (Figure 3C). None of the isolates expressed protease production, whereas all produced phospholipases. The Pop4 isolates exhibited high phospholipase activity, hemolytic activity, and gelatin hydrolysis. Most of them were high producers of urea and high or medium melanin producers.

It was observed that there was a difference in gelatin hydrolysis activity among Pop1 (0.77 ± 0.02) × Pop2 (0.97 ± 0.07), and Pop4 (0.66 ± 0.24) × Pop2 (0.97 ± 0.07) (*p* = 0.0014) (Figure 3E). All isolates of Pop1 showed low gelatin hydrolysis activity. All isolates of Pop4 showed high activity for phospholipases, hemolytic activity, and gelatin hydrolysis. In addition, most isolates of Pop4 were high producers of urea and high or medium producers of melanin.

### 3.3. Antifungal Susceptibility

The isolates exhibited susceptibility to ITZ, AMB, and KTZ and were shown to be wild-type by both EUCAST and CLSI methods (Table 4). When 16 µL/mL is considered a limit value for susceptibility dose-dependent (SDD) to FLZ for *Cryptococcus* spp., four (16.7%) isolates exhibited SDD (ST343, ST344, ST345, and ST347) (Figure 2A and Table 4). In accordance with the EUCAST criteria, four isolates (16.7%) exhibited resistance to FLZ (ST337, ST342, ST360, and ST361) (Table 4 and Figure 3). The agreement at ±two dilutions for both methods was 100% for ITZ, KTZ, and VRZ, 96% for AMB, and 92% for FLZ. The susceptibility profile of the EUCAST was similar among isolates in terms of their geographical origin. Differences in susceptibility profiles were presented for ITZ via CLSI among Pop3 (0.125± 0.0) × Pop4 (0.068 ± 0.04) (*p* = 0.047) (Figure 4H) and in Pop2 (0.33 ± 0.32) × Pop4 (0.037 ± 0.015) (*p* = 0.019) for AMB via EUCAST (Figure 4A).

### 3.4. The Minimum Spanning Tree

In order to infer patterns of evolutionary descent among clusters of related genotypes and to identify relations with virulence factors, antifungal susceptibility, clinical outcome, and geographic origin, the goeBURST analysis was applied (Figure 2). Two main clusters were identified. The first and minor are composed of the group founder (GF) GF353 and its descendants. The second and major are represented by GF338 and its descendants. All isolates from the TM region were grouped into GF338 as shown in Figure 2C. Regarding the production of virulence factors, the results show a correlation between gelatin hydrolysis absence and GF353 (Figure 2G). All isolates from GF345 presented high activity for phospholipases (Figure 2E) and urease (Figure 2D). Isolates from patients with poor outcomes were grouped in the GF344, whereas most of those who were grouped in the GF338 were cured (Figure 2F). Regarding antifungal susceptibility, most of the GF338 isolates were susceptible or wild-type to fluconazole by the CLSI method (Figure 2A), and the GF344 exhibited resistance to fluconazole by the EUCAST method (Figure 2B).

## 4. Discussion

The description of several outbreaks of cryptococcosis caused by the *C. deuterogattii* genotype in the last decades in some temperate areas of North America called the attention of the scientific community since it was formerly restricted to tropical and subtropical regions of the world [21,60]. Due to this fact, the evaluation of several aspects of this molecular type became more relevant [2,15,61,62].

In this sense, the identification of potential correlations between molecular profiles with geographic origin or phenotypic characteristics such as antifungal susceptibility, pathogenicity, and virulence factors, among others, is pivotal. Currently, these correlations are weak, and some discrepancies and questions remain to be solved [63,64,65,66,67].

Data on the production of capsular polysaccharide and melanin, urease, and phospholipase activity, among others, partially contributed to separating the *C. neoformans* species complex (CNSC) from the CGSC ones [1,21,68,69]. Although isolates from these species exhibit most of these features, it is unclear yet if different expression levels could have influenced the hypervirulence of *C. deuterogattii* isolates observed during the recent outbreaks [22].

In the present study, populations of *C. deuterogattii* isolates exhibited variable profiles regarding the production of virulence factors. The capsule size, urease production, and gelatine hydrolysis were more significant. In line with these findings, formerly reported studies described an association among STs and/or populations of *Cryptococcus* spp. with clinical and biological characteristics, such as patient outcome [63], male gender [66], immune response, capsule and melanin production [63], antifungal resistance [64,65], and genotype virulence [21]. The latter was evaluated by a *Galleria mellonella* experimental model, where it was pointed out that virulence is related to the distinct characteristics of individual strains and is not specific to a particular molecular type of CGSC [67].

A significant difference in urease activity among the populations evaluated was found. Urease is an enzyme that acts in phagosomal acidification by hydrolyzing urea to generate ammonia [70]. Its production has been considered pivotal during the process of central nervous system invasion [71,72]. Additionally, recently it was reported that urease acts together with melanization, which is another important fungal virulence factor [73].

The identification of specific genotypes of *Cryptococcus* spp. and their correlations with patterns of antifungal susceptibility and virulence could be an important epidemiological tool to improve vigilance for the emergence of resistant and/or more virulent strains [38,74].

Herein, the isolates from Pop4 exhibited a higher production of virulence factors than the others. This fact could indicate a greater virulence of these genetically more dispersed isolates in relation to other populations. Of note, it is important to mention that these isolates were not grouped in a monophyletic way like the other populations, and they exhibited the greatest genetic variability. Despite the small number of isolates included, the preliminary data obtained could suggest a relationship between genotypic differences in *C. deuterogattii* and higher expression of virulence factors and how this could influence the patient’s outcome.

Epidemiological surveillance of the antifungal resistance of *C. deuterogattii* isolates is pivotal in order to guide the therapy of patients with cryptococcosis caused by this genotype. Currently, validated antifungal breakpoints for *Cryptococcus* spp. are not available. Thus, the epidemiological cut-off values (ECVs) are used to aid in the detection of non-wild-type strains for antifungal susceptibility in a given fungal population [30,75]. Then, the cut-off point of 8 μg/mL has already been used in previous reports as a cut-off point for SSD to fluconazole [29,59,76]. All isolates herein evaluated presented a wild-type phenotype by the updated CLSI method, despite the fact that four presented SDD and four exhibited resistance by the EUCAST method using breakpoint interpretation for *Candida* species.

A decreased susceptibility among *C. deuterogattii* isolates with high MICs to azole drugs had already been observed by others elsewhere [35,36,77,78,79,80,81]. Several mechanisms of acquired resistance to fluconazole have been described. These include previous exposure [82], drug target alterations encoded by the gene ERG11 [83], subset populations presenting heterogeneous resistance that proliferate during treatment [83], the presence of group I introns in the mitochondrial large subunit rRNA gene (LSU) [27], and significantly higher expression of ABC transporters [28], among others.

The antifungal susceptibility difference among molecular types, STs, and/or populations had already been pointed out for the CNCS and CGCS complexes [64,65,84]. Antifungal susceptibility differences among populations to ITZ by CSLI and to AMB by EUCAST were herein identified as well. This fact is relevant since it raises the question of whether patients infected with these isolates could have different clinical outcomes.

The meaning of *Cryptococcus* spp. mixed infections in the clinical picture and outcome context of patients with cryptococcosis is unknown, and it needs to be better evaluated, mainly for *C. deuterogattii* [85,86,87]. In this report, *C. deuterogattii* isolates recovered from the same patient presented mixed infections (G8 and G9). They presented different STs (ST344 and ST345) and were recovered from the culture of two skin fragment samples collected at an interval of 22 days. The patient, a farmer who presented with primary cutaneous cryptococcosis after traumatism with eucalyptus tree logs, was apparently immunocompetent, and all clinical and laboratory exams were normal. It was cured after ten weeks of fluconazole. Despite his good outcome, these isolates presented differences related to the expression of virulence factors and antifungal susceptibility, which can raise the hypothesis that they represent a mixed infection and not a reinfection [88].

Data about mixed infections by *Cryptococcus* spp. are scarce, and this could be related to technical bias. Molecular studies usually select one single colony isolated from the culture of a unique anatomical site, which would make the identification of mixed infections unlikely [87,89]. Mixed infections with pathogenic species of *Cryptococcus* have already been reported elsewhere, using different molecular tools and with frequencies ranging from 16.7% to 66% [64,85,87,90,91]. The main fact supporting the occurrence of mixed infections could be the result of either co-inoculation or in vivo evolution (microevolution) [85]. However, microevolution could not explain the simultaneous presence of A and D serotype isolates in the same patient, as formerly described [85,90]. Despite the measures to avoid contamination in the routine mycology lab, cross-contamination of clinical samples cannot be ruled out.

Despite the low number of strains evaluated, the results herein described corroborate previous reports about the molecular diversity of the Brazilian VGII *C. gattii*/*deuterogatti* isolates and reinforce the occurrence of different virulence factors’ expression levels and antifungal susceptibility patterns among them. These facts are relevant in the clinical and outcome contexts of patients with cryptococcosis caused by this genotype.

## Figures and Tables

**Figure 1 jof-09-00889-f001:**
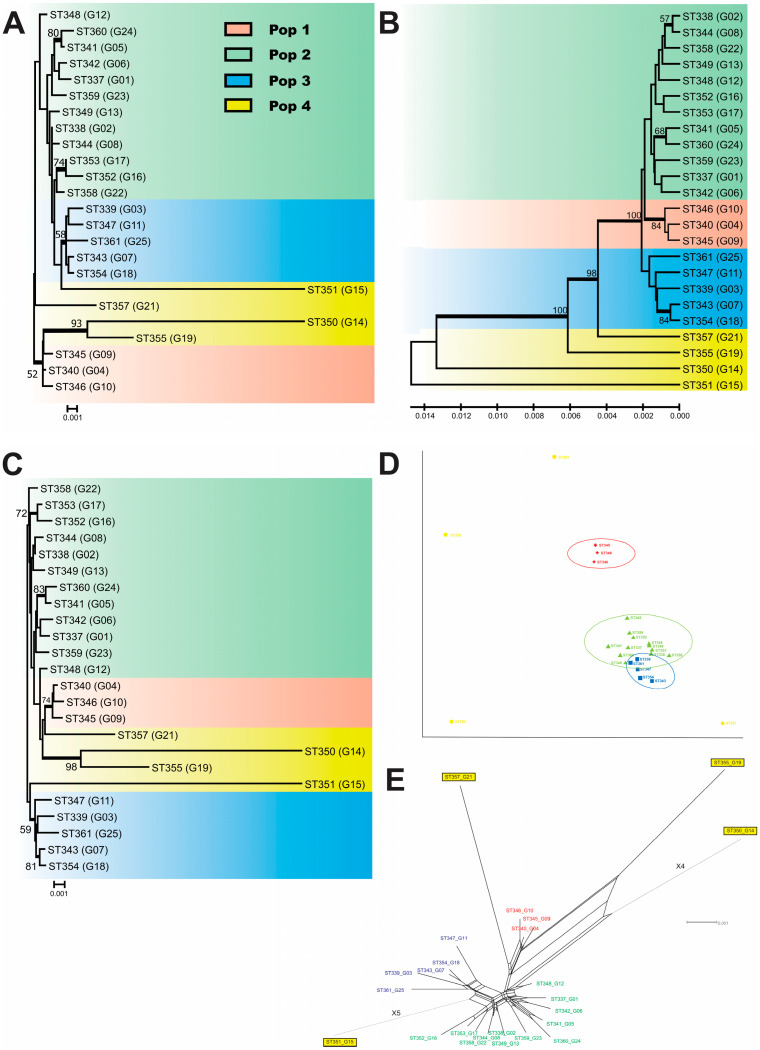
Phylogenetic analysis of 24 *Cryptococcus deuterogattii* isolates. Analysis performed by maximum likelihood (ML) (**A**), unweighted pair group method with arithmetic mean (UPGMA) (**B**), and neighbor-joining (NJ) (**C**) methods, using the concatenated data set of the seven MLST loci (*CAP59*, *GPD1*, *LAC1*, *PLB1*, *SOD1*, *URA5*, and the IGS1 region). The tree is drawn to scale, with branch lengths in the same units as those of the evolutionary distances used to infer the phylogenetic tree. The analysis involved 24 nucleotide sequences of isolates from the present study. There were a total of 4180 positions in the final dataset. Numbers at each branch indicate bootstrap values *>* 50% based on 1000 replicates by each of the three. Evolutionary analyses were conducted in MEGA 6. The main four groups are marked with five different colors. The isolates are described according to the sequence type number (ST), followed by your identification. (**D**) Classical Multidimensional Scaling Plot. The X and Y axes explain variabilities of 59.6% and 19.4%, respectively. Both axes explain 79% of the variability. (**E**) Split decomposition analysis applying the NeighborNet algorithm using the uncorrected-P parameter model and evidencing the diversity and branching ambiguities attributable to recombination events. The observation that isolates are linked to each other by multiple pathways and are forming an interconnected network rather than a single bifurcating tree is suggestive of recombination. The phi test for recombination implemented in the software SplitsTree showed significant evidence (*p <* 0.0001) for recombination. The STs belonging to the main clusters identified in the previous phylogenetic analysis were also separated using the split decomposition. The main clusters identified are highlighted as follows: Pop1: red; Pop2: green; Pop3: blue; and Pop4: yellow.

**Figure 2 jof-09-00889-f002:**
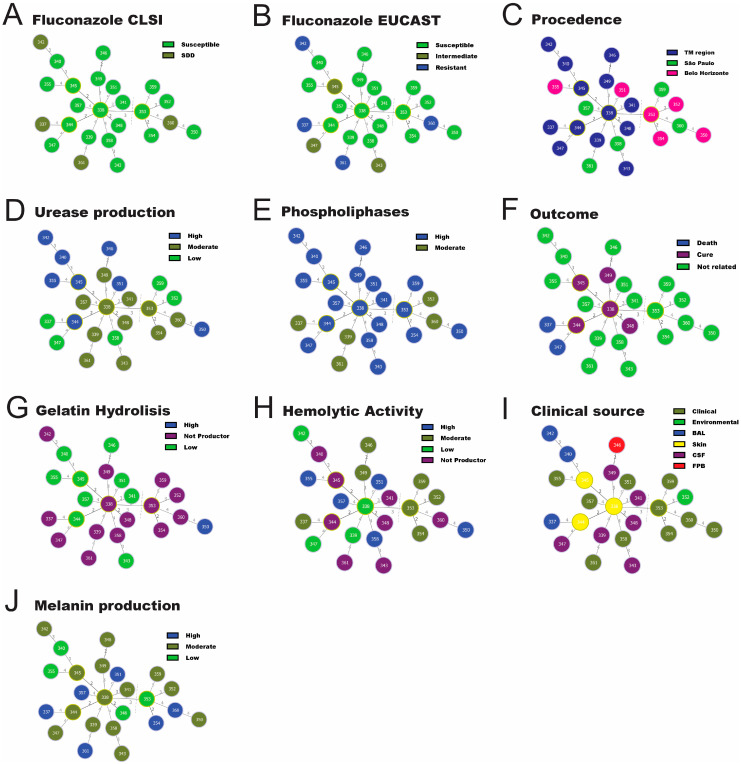
Minimum spanning tree of the 24 studied *Cryptococcus deuterogattii* isolates using the goeBURST algorithm. The distribution of the STs was compared with susceptibility to fluconazole by CLSI (**A**) and EUCAST (**B**) methods. Procedence (C). Production of the main virulence factors and clinical data are as follows: urease (**D**), phospholipase (**E**), outcome (**F**), gelatin hydrolysis (**G**), hemolytic activity (**H**), and clinical source (**I**) and melanin production (**J**). The group founders (GF) (highlighted by a yellow line) found were GF338; GF353; GF345; and GF344. Two main clusters were identified. The first and minor are composed of the GF353 and its descendants. The second major is represented by GF338 and its descendants. Each circle represents a unique ST. The numbers presented in the dashed branches represent at least one and the maximum of five differences in alleles, respectively. TM region—Triângulo Mineiro region.

**Figure 3 jof-09-00889-f003:**
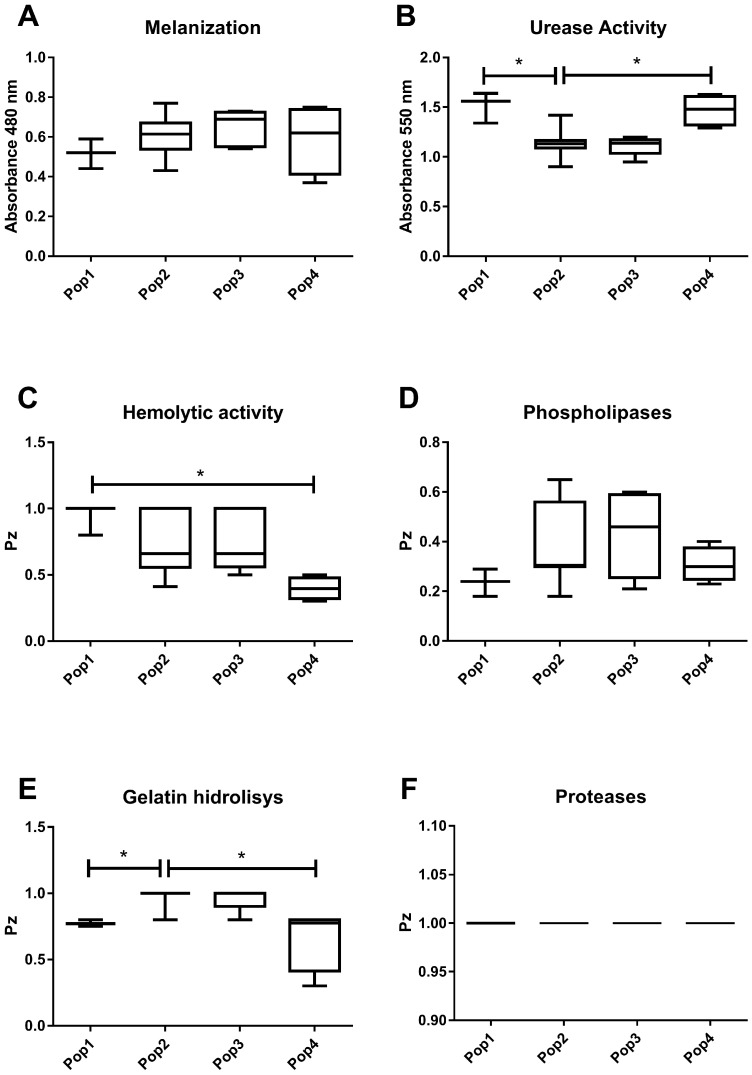
Comparison of virulence factors among the four populations of *Cryptococcus deuterogattii* evaluated in the present study. The virulence factors evaluated are melanization (**A**), urease activity (**B**), hemolytic activity (**C**), phospholipases (**D**), gelatin hydrolysis (**E**), and proteases (**F**). Statistically significant differences (*p <* 0.05) are marked with (*), followed by the Kruskal–Wallis test and Dunn’s test. The internal horizontal lines represent the median, the bars represent 25 ± 75% percentiles, and the horizontal lines represent the minimum and maximum.

**Figure 4 jof-09-00889-f004:**
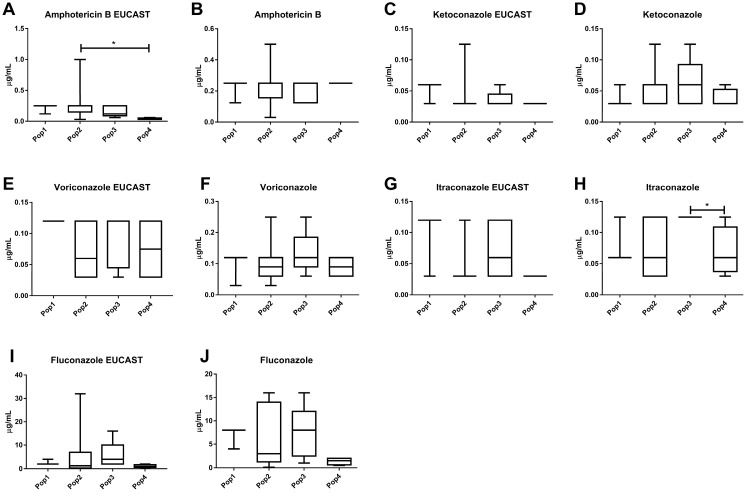
Comparison of the antifungal susceptibility patterns (μg/mL) among 24 *Cryptococcus deuterogattii* evaluated in the present study via Eucast and CLSI methods. The drugs evaluated are amphotericin B via EUCAST method (**A**), amphotericin B via CLSI method (**B**), ketoconazole via EUCAST method (**C**), ketoconazole via CLSI method (**D**), voriconazole via EUCAST method (**E**), voriconazole via CLSI method (**F**), itraconazole via EUCAST method (**G**), itraconazole via CLSI method (**H**), fluconazole via EUCAST method (**I**), and fluconazole via CLSI method (**J**). Statistically significant differences (*p <* 0.05) are marked with (*), followed by the Kruskal–Wallis test and Dunn’s test. The internal horizontal lines represent the median; the bars are 25 ± 75% percentiles, and the horizontal lines are 10 ± 90% percentiles. CLSI: Clinical and Laboratory Standards Institute. EUCAST: European Committee for Antimicrobial Susceptibility Testing.

**Table 1 jof-09-00889-t001:** DNA polymorphisms according to populations, biological sources, and places where *Cryptococcus deuterogattii* isolates were recovered.

Source (n)	Length	S	*Π*	K	h	Hd	D	FD	FF	FS	Theta-w	Rm	PHI
Total (24)	4180	282	0.00843	34.663	24	1.000	−2.218 *	−3.5908 *	−3.708 *	−6.599	75.516	13	<0.0001 *
Pop1 (3)	4156	9	0.00144	6.000	3	1.000	#	#	#	#	6.000	0	#
Pop2 (12)	4165	38	0.00269	11.182	12	1.000	−0.5062	−0.2678	−0.3766	−4.522	12.583	8	0.07459
Pop3 (5)	4165	27	0.00274	11.400	5	1.000	−0.8977	−0.8977	−0.9661	−0.167	12.960	1	1.0
Pop4 (4)	4178	253	0.03192	131.33	4	1.000	−0.6624	−0.6125	−0.6697	3.072	138.000	5	0.0004 *
BH (6)	4180	235	0.02070	85.400	6	1.000	−1.1961	−1.2008	−1.3194	1.673	102.920	6	0.0007 *
SP (5)	4163	58	0.00612	25.400	5	1.000	−0.6626	−06626	−0.7184	0.796	27.840	7	0.8632
TM (8)	4164	36	0.00317	13.179	8	1.000	−0.2710	−0.1162	−0.1700	−1.628	13.844	6	0.0142 *
Skin (3)	4164	12	0.00193	8.000	3	1.000	#	#	#	#	8.000	0	@
BAL (3)	4163	16	0.00257	10.667	3	1.000	#	#	#	#	10.667	0	@
CSF (6)	4164	33	0.00343	14.267	6	1.000	−0.0816	−01647	−0.1608	−0.440	14.453	5	0.0362 *

Legend: S—number of polymorphic sites; *π*—nucleotide diversity; K—average number of nucleotide differences; h—number of haplotypes; Hd—haplotype diversity; D—Tajima’s D; FD—Fu and Li’s D; FF—Fu and Li’s F; FS—Fu’s Fs; *—*p* < 0.05; Rm—minimum number of recombination events; Theta w—theta (per sequence) from S; Vtnr—variance of theta (no recombination); Vnfr—variance of theta (free recombination); θS—Watterson’s estimate per sequence; #—four or more sequences are needed to compute the tests. The DNA polymorphism was evaluated, excluding sites with gaps. The repeated sequence types from different regions are not included in the total number. PHI—Pairwise Homoplasy Index; BH—city of Belo Horizonte; SP—city of São Paulo; TM—Triângulo Mineiro region; BAL—bronchoalveolar lavage; CSF—cerebrospinal fluid; @—there are too few informative characters to use the Phi Test; n—number of isolates.

**Table 2 jof-09-00889-t002:** Genetic distance matrix (Fst) of *C. deuterogattii* populations.

	Pop1	Pop4	Pop3	Pop2
Pop1	0.00000			
Pop4	0.02372	0.00000		
Pop3	0.59959 *	0.12670 *	0.00000	
Pop2	0.49339 *	0.28040 *	0.24950 *	0.00000

Legend: *—*p* < 0.05.

**Table 3 jof-09-00889-t003:** Differences in sequences between isolates G8 and G9 of *Cryptococcus deuterogattii*.

*SOD1* Locus																		*GPD1* Locus	
AT	Position																	AT	Position
	19	20	21	22	23	24	25	97	387	396	430	435	527	535	550	637	707		157
AT14 (G8)	A	C	C	G	C	T	A	T	T	G	T	C	C	A	A	C	T	AT21 (G8)	G
AT58 (G9)	.	.	.	.	.	.	.	C	C	T	.	T	A	G	G	G	C	AT6 (G9)	.

Legend: AT—allele type.

**Table 4 jof-09-00889-t004:** Antifungal susceptibility of 24 *Cryptococcus deuterogattii* isolates by CLSI and EUCAST methods.

Antifungal	Method	MIC 50	MIC 90	Range	WT (%)	Isolates SDD or Intermediate (%)	Isolates Resistants (%)	MIC Geometric Mean
Amphotericim B	CLSI	0.12	0.25	0.03–0.5	100	NA	0	0.188
	EUCAST	0.12	0.25	0.03–1	NA	NA	NA	0.14
Ketoconazole	CLSI	0.06	0.03	0.03–0.12	NA	NA	0	0.04
	EUCAST	0.03	0.06	0.03–1	NA	NA	NA	0.03
Itraconazole	CLSI	0.06	0.12	0.03–0.12	100	NA	0	0.06
	EUCAST	0.03	0.06	0.03–0.06	NA	NA	NA	0.03
Fluconazole	CLSI	2	16	0.12–16	100	4 (16.7)	0	2.78
	EUCAST	2	8	0.12–32	NA	11 (45.8)	4 (16.7)	1.64
Voriconazole	CLSI	0.12	0.25	0.03–0.25	100	NA	0	0.08
	EUCAST	0.06	0.12	0.03–0.12	NA	NA	NA	0.06

Legend: MIC—minimum inhibitory concentrations; NA—not applied; MIC50 and MIC90—MICs at which 50% and 90% of isolates were inhibited, respectively.

## Data Availability

The sequencing data used were retrieved from the GenBank database available at the National Center for Biotechnology Information (NCBI).

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
