# Peer review of "Genotyping Analysis of Cryptococcus deuterogattii and Correlation with Virulence Factors and Antifungal Susceptibility by the Clinical and Laboratory Standards Institute and the European Committee on Antifungal Susceptibility Testing Methods"

_jof, 2023, doi:10.3390/jof9090889_

Round 1

Reviewer 1 Report

This study showed the molecular diversity of the Brazilian Cryptococcus deuterogattii (VGII) isolates which can be divided into four genotype groups (Pop1-4) and the isolates from Pop4 exhibited a higher production of virulence factors than the others. They also found their significant correlations between patterns of antifungal susceptibility to itraconazole and amphotericin B and virulence factors expression.  And in an uncommon case of primary cutaneous cryptococcosis, two isolates of Cryptococcus deuterogattii with different STs (ST344 and ST345) were recovered from the same patient with an interval of 22 days, that supported the exist of Cryptococcus deuterogattii mixed infection, not microevolution. A hard work has done for this study and the results indicated the value for further identification the relationship among the genotype and virulence and its relevant outcome with more clinical isolates and animal models. Here, a few questions as follows for the further considerations.

11.  Line 38: Cryptococcus neoformans complex species (CNCS) is commonly defined as Cryptococcus neoformans species complex (CNSC). The same as Cryptococcus gattii species complex (CGSC).

12.  I would appreciate more detailed clinical information such infection sites, CrAg titer, patient immune status and outcome are included for these patients with cryptococcosis.  It would be great if the significant relationship can be found between the clinical features and genotyping or virulence.

23.  From the MIC results can see that there is a little difference between assay, CLSI and EUCAST. Would you please give us a further explanation or guidance in our clinical practice.

Author Response

Reviewer 1

This study showed the molecular diversity of the Brazilian Cryptococcus deuterogattii (VGII) isolates which can be divided into four genotype groups (Pop1-4) and the isolates from Pop4 exhibited a higher production of virulence factors than the others. They also found their significant correlations between patterns of antifungal susceptibility to itraconazole and amphotericin B and virulence factors expression.  And in an uncommon case of primary cutaneous cryptococcosis, two isolates of Cryptococcus deuterogattii with different STs (ST344 and ST345) were recovered from the same patient with an interval of 22 days, that supported the exist of Cryptococcus deuterogattii mixed infection, not microevolution. A hard work has done for this study and the results indicated the value for further identification the relationship among the genotype and virulence and its relevant outcome with more clinical isolates and animal models. Here, a few questions as follows for the further considerations.

Point 1.  Line 38: Cryptococcus neoformans complex species (CNCS) is commonly defined as Cryptococcus neoformans species complex (CNSC). The same as Cryptococcus gattii species complex (CGSC).

Response 1: The changes were carried out throughout the manuscript.

Point 2.  I would appreciate more detailed clinical information such infection sites, CrAg titer, patient immune status and outcome are included for these patients with cryptococcosis.  It would be great if the significant relationship can be found between the clinical features and genotyping or virulence.

Response 2: The main clinical information can be found in S1 table. A sentence describing this was added to manuscript (lines 94-98). The isolates came from different regions from Brazil and it was very difficult the access to the clinical data

Point 3.  From the MIC results can see that there is a little difference between assay, CLSI and EUCAST. Would you please give us a further explanation or guidance in our clinical practice.

Response 3: These results are in line with different authors have shown a high correlation between the two protocols for azoles and amphotericin B for both Candida and Cryptococcus isolates (1. Delma FZ, et al., 2020 Comparison of MIC Test Strip and Sensititre YeastOne with the CLSI and EUCAST Broth Microdilution Reference Methods for In Vitro Antifungal Susceptibility Testing of Cryptococcus neoformans. Antimicrob Agents Chemother; 2. Ceballos-Garzon A, et al., 2022 Head-to-head comparison of CLSI, EUCAST, Etest and VITEK®2 results for Candida auris susceptibility testing. Int J Antimicrob Agents; 3. Cuenca-Estrella M, et al. 2010, Comparison of the Vitek 2 antifungal susceptibility system with the clinical and laboratory standards institute (CLSI) and European Committee on Antimicrobial Susceptibility Testing (EUCAST) Broth Microdilution Reference Methods and with the Sensititre YeastOne and Etest techniques for in vitro detection of antifungal resistance in yeast isolates. J Clin Microbiol; 4. Pfaller MA et al., 2014 Comparison of EUCAST and CLSI broth microdilution methods for the susceptibility testing of 10 systemically active antifungal agents when tested against Candida spp. Diagn Microbiol Infect Dis). However, to our knowledge, there are no studies comparing the two methodologies in C. deuterogatti. In the present study, we also found a strong correlation between the two protocols. Thus, both methods can be used in C. deuterogattii to assess antifungal susceptibility

Reviewer 2 Report

Dear authors

Thank you for submitting your draft titled "Genotyping analysis of Cryptococcus deuterogattii and correlation with virulence factors and antifungal susceptibility by CLSI and EUCAST methods" . I appreciate your effort and  I am grateful to submit my comments and suggest changes to improve the paper. I request only minor clarifications in the remarks that I add in the draft in the attached .pdf file.

Author Response

Reviewer 2.

Thank you for submitting your draft titled "Genotyping analysis of Cryptococcus deuterogattii and correlation with virulence factors and antifungal susceptibility by CLSI and EUCAST methods" . I appreciate your effort and  I am grateful to submit my comments and suggest changes to improve the paper. I request only minor clarifications in the remarks that I add in the draft in the attached .pdf file.

Point 4. Lines 77-78. Registred?

Response 4: Sorry, we didn't fully understood the question. To our knowledge both protocols are standardized by CLSI (document M27-A3) and EUCAST (document EDef 7.1).

Point 5. Lines 277-278. Differences in urease activity among Pop1 (1.513 ± 0.15) × Pop2 (1.135 ± 0.11) and in 277 Pop4 (1.47 ± 0.14) × Pop2 (1.135 ± 0.11) (p=0.0042) were found (Fig. 3B). “This data seems very relevant to me, however it is not commented extensively in the discussion and it is given little importance in the discussion... I recommend discussing this finding more extensively and mentioning it in your conclusions”

Response 5: A paragraph has been added to the discussion about urease production (lines 401-409).

Pointe 6. Lines 399-400. In this report, C. deuterogattii isolates recovered from the same patient presented mixed infections (G8 and G9). “there is a possibility of cross contamination in the lab... do you have evidence that this isolate is different from other strains of the same variety handled in your laboratory?”

Response 6: We believe which this possibility is remote. In the lab routine are take several precautions to prevent cross-contamination of samples, including: exchange of tips during manipulation of isolates, opening of cryotubes and DNA tubes using disposable paper, among others. In addition, the sequence types of the referred isolates are unique, that is, they were only found in these isolates

Pointe 7. Lines 415-416. However, microevolution could not explain the simultaneous presence of A and D sorotype isolates in the same patient, as formerly described [81,86]. “Have you considered the possibility of cross-contamination of clinical samples if they are being processed in the same laboratory? Add some comments related to this possibility in the discussion...”

Response 7: A sentence has been added to the discussion about this (lines 463-464).

Pointe 8. Line 416. presence of A and D sorotype isolates. “serotype?”

Response 8: the word was corrected (line 462).
